# Physicochemical Properties of Chicken Breast and Thigh as Affected by Sous-Vide Cooking Conditions

**DOI:** 10.3390/foods12132592

**Published:** 2023-07-03

**Authors:** Sin-Woo Noh, Dong-Heon Song, Youn-Kyung Ham, Na-Eun Yang, Hyun-Wook Kim

**Affiliations:** 1Department of Animal Science & Biotechnology, Gyeongsang National University, Jinju 52725, Republic of Korea; sinwoonoh@naver.com (S.-W.N.); timesoul@naver.com (D.-H.S.); 2Department of Animal Science, Sangji University, Wonju 26339, Republic of Korea; kham21@sangji.ac.kr; 3Department of GreenBio Science, Gyeongsang National University, Jinju 52725, Republic of Korea; ly2223@naver.com

**Keywords:** collagen content, myofibrillar fragmentation index, tenderness, water-holding capacity

## Abstract

Sous-vide is a cooking method used to improve the tenderness and juiciness of chicken breast. However, the comparative changes in meat quality attributes of sous-vide cooked chicken breast and thigh muscles are not fully understood. The objective of this study was to investigate the effects of sous-vide cooking conditions, based on collagen denaturation temperature of intramuscular connective tissue, on the physicochemical properties of chicken breasts and thighs. Chicken breast and thigh were cooked at four sous-vide cooking conditions (55 °C for 3/6 h and 65 °C for 3/6 h) and conventional cooking at 75 °C (core temperature of 71 °C) as control. No significant differences in pH and lightness were found between the sous-vide cooking conditions. Moisture content, cooking loss, protein solubility, shear force, myofibrillar fragmentation index, and lipid oxidation were affected by sous-vide cooking conditions (*p* < 0.05). The decreased shear force and total collagen content of 65 °C sous-vide cooking treatment might be associated with collagen denaturation (*p* < 0.05). Sous-vide cooking at 55 °C could decrease cooking loss, with higher moisture than sous-vide cooking at 65 °C (*p* < 0.05). These tendencies on water-holding capacity and shear force at the two different temperatures were similarly observed for both chicken breast and thigh. Therefore, this study indicates that chicken breast and thigh are similarly affected by the sous-vide cooking conditions and suggests that a novel strategy to apply together two temperature ranges based on the thermal denaturation of intramuscular connective tissue would be required.

## 1. Introduction

Sous-vide cooking is generally known as a culinary technique to heat vacuum-packaged food under a low controlled temperature for a long time [1]. In the meat service sector, this cooking method has been extensively used to serve tender and juicier meat. Moreover, some previous studies have found additional positive impacts of sous-vide cooking, such as a reduction in nutrient loss and prevention of the loss of flavor-related volatile compounds during meat cooking [2,3]. For these reasons, sous-vide cooking is now considered a promising thermal process for improving the palatability of processed meat products, mainly tenderness, juiciness, and flavor. Moreover, despite the relatively low cooking temperature, it has been reported that sous-vide cooking may have no adverse impacts on the microbial safety of meat during storage [4].

In practice, sous-vide cooking improves the tenderness and juiciness of chicken breast and provides an unexpected crumbly and dry mouthfeel [1,3]. However, the sous-vide conditions for guaranteeing the desirable positive effects on chicken breasts have varied, particularly regarding temperature and time. Park et al. [1] found that sous-vide cooking at 60 °C for 120–180 min (experimental setting under 60–70 °C for 60–180 min) could improve the palatability of chicken breast. However, Haghighi et al. [4] suggested that sous-vide cooking at 60 °C for 60 min could be an optimal condition to improve the eating quality attributes of the chicken breast. Furthermore, Hasani et al. [5] recently reported that a temperature combination of 50 and 60 °C for sous-vide cooking could improve the cooking loss and gumminess of chicken breast. Thus, sous-vide cooking at 60 °C might be optimal for guaranteeing desirable quality attributes of sous-vide cooked meat.

During the cooking process of meat, muscle proteins related to water-holding capacity and tenderness are thermally denatured at different temperatures; myofibrillar and stromal proteins are 40–50 °C and 65–75 °C, respectively [5]. The biochemical changes cause water loss from muscle tissue and shrinkage of muscle structure, resulting in a decline in juiciness and tenderness. Moreover, thermal shrinkage of collagenous connective tissues changes the structural integrity, which also affects the juiciness and tenderness of cooked meat. According to Purslow [6], the thermal denaturation of collagen in intramuscular connective tissues could occur at 55–60 °C at a slow heating rate, which might be related to the effects of low temperature and longtime cooking. The connective tissue of chicken muscle is mostly intramuscular; thus, it could be expected that temperature-inducing intramuscular collagen denaturation is one significant factor related to the efficacies of sous-vide cooking. However, there have been no experimental approaches to determine sous-vide conditions based on the thermal denaturation of intramuscular collagen.

The impacts of thermal processing on meat quality attributes differ depending on intrinsic factors, mainly muscle fiber composition [7]. Chicken breasts and thighs have different muscle fiber characteristics called ‘white muscle’ and ‘red muscle’, respectively. However, most previous studies regarding the efficacies of sous-vide cooking on chicken meat quality have focused on improving the sensory acceptance of chicken breasts. Even though chicken breast and thigh meat are often cooked together, there is no available literature on the meat quality attributes of chicken breast and thigh cooked under the same sous-vide cooking conditions. Therefore, the objective of this study was to investigate the implications of sous-vide cooking conditions based on the denaturation temperature of intramuscular connective tissue on the meat quality attributes of chicken breast and thigh muscles.

## 2. Materials and Methods

### 2.1. Raw Materials

Forty-five raw chicken carcasses (commercial broilers, 1354 ± 44.86 g average live weight), which were fed commercial crumble-pellet feeds and reared in a multilayer cage system farm, were purchased from a local poultry processing company and were used for the experiment within 1 day after purchase. Fifteen chicken carcasses were randomly grouped as a batch, from which thirty chicken breasts and thighs were manually collected and randomly assigned to five cooking condition treatments (six breasts and thighs/treatment/batch).

### 2.2. Sous-Vide Cooking Procedure

The skinless chicken breasts and thighs were individually vacuum-packaged in a nylon/polyethylene (PA/PE) pouch (150 cm × 200 cm, thickness 65 μm). Five cooking conditions based on the thermal denaturation of intramuscular connective tissues were performed in a constant-temperature and circulation water bath. The temperature and time settings were as follows: conventional cooking at 75 °C until 71 °C core temperature (as control); sous-vide cooking at 55 °C for 3 h (55 °C/3 h); 55 °C for 6 h (55 °C/6 h); 65 °C for 3 h (65 °C/3 h); and 65 °C for 6 h (65 °C/6 h). The core temperature of the control group was monitored using an insert-type digital thermometer (Tes-1305, Tes Electrical Corp., Taipai, Taiwan) with an iron constantan thermocouple, and the control samples were removed from the water bath when the core temperature reached 71 °C (average heating rate of 8 °C/min). All cooked samples were cooled at room temperature (20 ± 1 °C) for 3 h and used for further analysis.

### 2.3. Analysis of Sous-Vide Cooked Chicken Breast and Thigh

#### 2.3.1. pH Measurement

The pH of the samples was measured in triplicates using an insert-type pH meter (HI99163, Hanna Instruments, Smithfield, RI, USA. The pH meter was calibrated using the manufacturer’s standard buffer solutions (pH 4.01 and 7.00, Thermo Fisher Scientific, Waltham, MA, USA).

#### 2.3.2. Moisture Content

The moisture content of cooked samples was measured in triplicate according to the oven-drying method (950.45B) of the Association of Official Analytical Chemists [8]. Moisture content was expressed as g of moisture per 100 g of sample (g/100 g).

#### 2.3.3. Total Collagen Content

The total collagen content of the sample was determined in duplicate according to the method described by Starkey et al. [9] with slight modification. A total of 100 milligrams of the sample were mixed with 3 mL of 3.5 M sulfuric acid and hydrolyzed at a 105 °C drying oven for 16 h. The hydrolysate was diluted with 50 mL of distilled water and then filtered through a filter paper (Whatman no. 1). Then, 1 milliliter of the filtrate was mixed with 3.75 mL of distilled water and 0.25 mL of 1.5 M NaOH, and 0.5 mL of the mixture was finally reacted with 0.25 mL of oxidant solution (50 mM chloramine-T hydrate, 156 mM citric acid, 375 mM NaOH, 661 mM sodium acetate trihydrate, 29% *v*/*v* 1-propanol, pH 6.0) for 20 min. After the reaction, the reactant was mixed with 0.25 mL of color-developing reagent (246 mM 4-dimethylaminobenzoaldehyde, 35% *v*/*v* perchloric acid, 65% *v*/*v* 2-propanol) and heated in a 60 °C water bath for 15 min. After cooling for 3 min, the absorbance of the mixture was read at 558 nm, and total collagen content was determined using a standard curve (hydroxyproline solution of 0.6, 1.2, 1.8, and 2.4 μg/mL). Total collagen content was calculated using the following equation: Total collagen (mg/g dry matter) = hydroxyproline (μg/mL) × 7.25/1000/(sample weight/250).

#### 2.3.4. Instrumental Color Measurement

The instrumental color of the bone side surface on cooked samples was evaluated using a colorimeter (Chroma meter, CR 400, Minolta, Osaka, Japan) with an illuminant C and a 2° standard observer. The colorimeter was calibrated with a standard white plate (CIE L* = +93.01, CIE a* = −0.25, CIE b* = +3.50). Values for CIE L* (lightness), CIE a* (redness), and CIE b* (yellowness) of each sample were measured at six random locations, and the values were averaged [10].

#### 2.3.5. Cooking Loss

Cooking loss was expressed as the percentage weight difference between raw and cooked samples using the following equation: cooking loss (%) = (the weight before cooking (g) − the weight after cooking (g)/the weight before cooking (g) × 100.

#### 2.3.6. Total Protein Solubility

The total protein solubility of the samples was measured in triplicate according to the method of Warner et al. [11]. The protein concentration of the sample was measured using the Biuret method [12], and protein solubility was expressed as mg of soluble protein per g of sample (mg/g).

#### 2.3.7. Myofibrillar Fragmentation Index

Myofibrillar fragmentation index (MFI) was measured in triplicate using the method of Olson and Stromer [13] with slight modification. Four grams of the sample were homogenized with 40 mL of isolation buffer (100 mM KCl, 20 mM K-phosphate, pH 7.0, 1 mM NaN_3_) for 30 s, then centrifuged at 1000× *g* for 10 min and suspended at the residues again with 20 mL of isolation buffer. Then, the same process was performed twice under the same conditions, and connective tissue was removed by filtering with a polyethylene sieve of 18 mesh scale and then further centrifuged. After centrifugation, the myofibrillar sediment was suspended again with 20 mL of isolation buffer, and the protein concentration of the sediment suspension was measured using the Biuret method. The protein concentration of myofibrillar sediment was diluted at 0.5 ± 0.05 mg/mL using the isolation buffer, and the absorbance was measured at 540 nm. MFI was expressed by multiplying the absorbance value by 200.

#### 2.3.8. Shear Force

For shear force measurement, at least four rectangular parallelepiped shape samples (3.0 cm × 1.0 cm × 1.0 cm) were obtained from the middle portion of each cooked chicken breast or from the *M. iliotibialis lateralis* in the thigh meat. The shear force was measured using a texture analyzer (CT3 50K, AMETEK Brookfield, Middleboro, MA, USA) equipped with a V-shape Warner–Bratzler shear force blade. The experimental condition was set to a maximum load of 50 g and a test speed of 2 mm/s [14].

#### 2.3.9. Lipid Oxidation

Lipid oxidation was determined in triplicate according to the 2-thiobarbituric acid reactive substance (TBARS) method described by Buege and Aust [15] with minor modification. Five grams of sample was mixed with 15 mL of distilled water and 100 µL of 6% 2,6-di-tert-butyl-4-methylphenol (BHT) and homogenized at 12,000 rpm for 15 s. Two milliliters of the homogenate were reacted with 4 mL of TBA solution (20 mM 2-thiobarbituric acid in 15% trichloroacetic acid). The mixture was heated at 80 °C for 15 min in a water bath and cooled in cold water for 10 min. After cooling, the sample was mixed and then centrifuged at 2000× *g* (20 °C) for 10 min (Combi-514R, Hanil SME, Anyang, Republic of Korea). The supernatant was filtered through a filter paper (Whatman no. 4), and the absorbance of the filtrate was measured at 531 nm using a spectrophotometer (Libra S22, Biochrome, Cambridge, UK). The TBARS value was calculated using the following equation and expressed as mg of malondialdehyde per kg of the sample (mg MDA/kg sample): TBARS value (mg MDA/kg sample) = 5.3102 × absorbance 531 nm + 0.0356.

### 2.4. Statistical Analysis

An analysis of variance (ANOVA) was conducted on the measured variables using the two-way ANOVA procedure of the SPSS program (SPSS Inc., Chicago, IL, USA), in which the main effects were considered sous-vide cooking conditions, muscle type, and their interactions. Duncan’s multiple range test was used to determine significant differences between means (*p* < 0.05).

## 3. Results and Discussion

Chicken breasts and thighs clearly present different muscle characteristics due to their muscle fiber composition [16,17]. Many previous studies have extensively compared the meat quality attributes of chicken breasts and thighs [18,19]. Therefore, this study focused on sous-vide cooking conditions and interaction effects on chicken breasts and thighs.

### 3.1. pH and Color Characteristics of Sous-Vide Cooked Chicken Breast and Thigh

The pH and color of chicken breast and thigh with different sous-vide cooking conditions (at 55/65 °C for 3/6 h) are shown in Table 1. No interaction between muscle type and sous-vide condition on pH value was found (*p* > 0.05). The pH values of cooked chicken breast and thigh were 6.07 and 6.37, respectively (*p* < 0.05). However, the pH values were unaffected by the different sous-vide cooking conditions (*p* > 0.05).

A significant interaction between sous-vide cooking condition and muscle type effect was confirmed only in CIE b* (yellowness) but not in L* (lightness) and a* (redness). The cooked chicken breast showed a higher lightness (*p* < 0.05) but lower redness (*p* < 0.05) and yellowness (*p* < 0.05) as compared to cooked chicken thigh. Moreover, the redness and yellowness of cooked chicken breast and thigh were affected by the different sous-vide cooking conditions (*p* < 0.001). During the thermal process, the denaturation of myoglobin, a pigment-protein, begins at 55–65 °C, and as the temperature rises, more denaturation progresses [20]. Thus, the increased temperature of sous-vide cooking decreases the redness of cooked chicken meat, which was similar to the results of previous studies on sous-vide cooked pork, beef, and goat meat [21,22]. In addition, Park et al. [1] reported that sous-vide cooking at 70 °C for 1 h resulted in lower redness of chicken breast compared to sous-vide cooking at 60 °C for 3 h.

In general, oxymyoglobin is denatured at 75–80 °C, causing browning, and thigh muscle is known to have higher myoglobin content than breast muscle [23]. In the previous study, the browning due to heating in thigh muscle with a high concentration of oxymyoglobin progressed further, and there was a significant difference in yellowness [23]. Therefore, this result showed that the increased temperature and time of sous-vide cooking could decrease redness but increase the yellowness of chicken meat, in which the yellowness of chicken thigh meat was more sensitive to the change in the sous-vide cooking conditions than chicken breast.

### 3.2. Cooking Loss and Moisture Content of Sous-Vide Cooked Chicken Breast and Thigh

Cooking loss and moisture content of the chicken breast and thigh muscles with different sous-vide cooking conditions are shown in Figure 1 and Table 2, respectively. No significant interactions between muscle type and sous-vide cooking conditions on cooking loss and moisture content were found (*p* > 0.05). Chicken breast and thigh showed similar cooking loss and moisture content (*p* > 0.05). The lowest cooking loss (9.79%) of chicken meat was observed for sous-vide cooking at 55 °C for 3 h, but sous-vide cooking at 65 °C for 6 h caused the highest cooking loss (*p* < 0.05). Moreover, the cooking loss of chicken muscles increased as the cooking temperature and time increased. As a result, the lowest moisture content of chicken meat was found in sous-vide cooking at 65 °C for 6 h (*p* < 0.05; Table 2). Thus, the moisture content of chicken muscles tended to show opposite changes against cooking loss.

One of the major factors affecting the water-holding capacity in cooked meat is the shrinkage of the myofibrillar protein. Eighty percent of the moisture in the meat is held by myofibrillar proteins [24]. When myofibrillar protein is thermally denatured, it shrinks transversely at 40–60 °C, thus extending the space between myofibrillar, but at 60–65 °C, longitudinal shrinkage occurs, causing significant moisture loss [24]. According to Park et al. [1], the cooking loss of sous-vide chicken breast cooked at 70 °C was significantly higher than sous-vide cooking at 60 °C, and the prolonged cooking time at the same temperature markedly increased the cooking loss of sous-vide chicken breast. Thus, it seems that the increased cooking loss at 65 °C sous-vide chicken meat with decreasing moisture content was likely due to the more severe shrinkage of muscle protein.

### 3.3. Total Collagen Content of Sous-Vide Cooked Chicken Breast and Thigh

The total collagen content of sous-vide cooked chicken breast and thigh muscles is shown in Table 2. The total collagen content was significantly different by muscle types and sous-vide cooking conditions (*p* < 0.05), but there were no significant interactions between muscle types and sous-vide cooking conditions (*p* > 0.05). One of the major changes in the heat treatment of meat is decreasing shear force by the solubilization of collagen, and in particular, the heating temperature is a major factor in the decrease in hardness caused by collagen [25]. The denaturation temperature of collagen was reported from 53 °C to 63 °C [26]. Moreover, according to Purslow [6], cooking at a slow heating rate, such as sous-vide heating, can cause denaturation of collagen between 55 and 60 °C. In this study, the collagen content of sous-vide chicken also showed a significant decrease in the 65 °C, 3–6 h treatment than control, 55 °C 3 h treatment, resulting in collagen denaturation and solubilization, resulting in a decrease in the residual collagen content.

### 3.4. Total Protein Solubility of Sous-Vide Cooked Chicken Breast and Thigh

The total protein solubility of sous-vide cooked chicken breast and thigh is shown in Table 2. No significant interaction between muscle type and sous-vide cooking conditions was found, and similar protein solubility was observed for both chicken breast and thigh (*p* > 0.05). The total protein solubility of chicken meat was highest in sous-vide cooking at 55 °C (*p* < 0.05); however, an increase in cooking temperature to 65 °C caused no difference in the total protein solubility when compared to the control group (*p* > 0.05). Protein solubility has been used as an indicator of the degree of protein denaturation, and protein aggregation due to thermal denaturation decreases protein solubility [27]. Protein denaturation is divided into two stages according to temperature. Most sarcoplasmic protein denaturation occurs at 40–60 °C, and myofibrillar protein denaturation occurs at 70–90 °C [7,27]. In this study, the protein solubility of chicken meat decreased as the cooking temperature increased. This observation was consistent with the results of previous studies [28,29]. Thus, it seems that sous-vide cooking at 65 °C might have a greater impact on the denaturation of myofibrillar proteins than that at 55 °C, while both temperatures are below the thermal denaturation temperature already known. According to Ayub and Ahmad [30], myosin, a major myofibrillar protein, causes extensive denaturation when heated at 55 °C for 3 h. On the other hand, Martens et al. [31] reported that actin could be denatured under cooking at 65 °C for 3 h. Thus, the reason for similar protein solubility between 3 and 6 h at the same cooking temperature was likely that sous-vide cooking for 3 h might be enough to denature myofibrillar protein at each cooking temperature.

### 3.5. Myofibrillar Fragmentation Index (MFI) of Sous-Vide Cooked Chicken Breast and Thigh

The MFI of sous-vide cooked chicken breast and thigh was determined to compare the difference in structural change (Table 2). The MFI of sous-vide cooked chicken was affected by muscle types (*p* < 0.05), sous-vide cooking conditions (*p* = 0.002), and their interaction (*p* = 0.003). The MFI of the control group was significantly lower than sous-vide cooking treatments (*p* < 0.05). As the cooking temperature increased from 55 to 65 °C, the MFI of cooked chicken meat numerically decreased (*p* > 0.05). However, the change in MFI due to the increased cooking temperature was observed for only chicken breast (*p* < 0.05), but the different sous-vide cooking conditions had no impact on the MFI of chicken thigh (*p* > 0.05). Changes in the MFI of chicken breast are mainly associated with actomyosin dissociation and Z-disc destruction. Previously, Wang et al. [32] reported that actomyosin dissociation in duck breast increased significantly at 50 °C, peaked at 60 °C, and decreased rapidly at 80 °C. In addition, Okitani et al. [33] reported that the dissociation of actomyosin showed a high level for up to 1 h when cooked at 60 °C, and the dissociated actin was insolubilized at 65 °C of cooking temperature. However, there was no significant difference in MFI according to the treatment group in the thigh muscle, which is presumed by other factors. It has been reported that in the early stage of heating, inosine 5′-monophosphate (IMP) or adenosine-5′-monophosphate (AMP) dissociates actomyosin into actin and myosin more markedly at 60–65 °C than at 80 °C [33]. MFI may be affected by the content of AMP and IMP remaining in the muscle during heating. According to Jayasena et al. [34], the AMP content of chicken thigh muscle was more than twice as low as that of chicken breast muscle (*p* < 0.05), and the content of IMP was not significantly different (*p* > 0.05). Thus, the difference in the MFI of the thigh muscle was not observed despite the change in heating condition because the AMP content of the thigh muscle was low, and the dissociation of actomyosin was not large enough.

### 3.6. Shear Force of Sous-Vide Cooked Chicken Breast and Thigh

The shear force of sous-vide cooked chicken breast and thigh muscles is shown in Figure 2. No significant interaction between muscle types and sous-vide cooking conditions on the shear force was found (*p* > 0.05). Sous-vide cooking at 65 °C could decrease the shear force of chicken meat (*p* < 0.05), but sous-vide cooking at 55 °C had no impact on the shear force (*p* > 0.05). In addition, there was no significant difference in the shear force due to cooking time within the same temperature (*p* > 0.05). Thermal treatment of meat causes several changes, including myofibrillar protein denaturation and shrinkage, which contribute to the increased shear force of cooked meat [27]. In general, both cooking temperature and time affect the tenderness of meat, but the cooking temperature has a greater effect on myofibrillar protein shrinkage than cooking time [26]. According to Barbanti and Pasquini [35], the decrease in shear force during cooking was mainly due to the solubilization of the connective tissue, whereas denaturation of myofibrillar protein led to an increase in shear force. Therefore, the shear force increase of 55 °C treatment in chicken breast and thigh muscles was caused by the denaturation of myofibrillar protein. However, the causes of shear force reduction in 65 °C treatment of chicken breast and thigh muscles are estimated to be caused by different factors. The decrease in the shear force of meat is mainly due to the solubilization of intramuscular connective tissue such as collagen. In this study, the collagen content of sous-vides chicken muscles was significantly lower at 65 °C 6 h, which is consistent with the shear force reduction trend of 65 °C treatment. Also, according to the report by Baldwin [24], a slow change caused by sous-vide cooking dissolves collagen, denatures it into gelatin, and improves tenderness by reducing adhesion between muscle fibers. Regarding the difference in shear force between chicken breast and thigh, it is well known that the difference in meat quality between chicken breast and thigh meat is due to muscle fiber composition, but the results of instrumental tenderness differences were not consistent in previous studies [36,37]. This may be due to postmortem time and cooking methods [37].

### 3.7. Lipid Oxidation of Sous-Vide Cooked Chicken Breast and Thigh

The lipid oxidation stability of sous-vide cooked chicken breast and thigh were determined with a TBARS assay (Figure 3). No significant interaction between muscle types and sous-vide cooking conditions on TBARS value was found (*p* > 0.05). Sous-vide cooking at 55 °C resulted in the lowest TBARS values (*p* < 0.05). In each muscle, an increase in sous-vide cooking temperature increased the TBARS value (*p* < 0.05), and different cooking times had no impact on the TBARS value of chicken breast (*p* > 0.05). The change in TBARS values due to different cooking temperatures may be related to the inactivation of endogenous antioxidant enzymes in the meat or the formation of antioxidative compounds. Mei et al. [38] reported that heating temperatures above 60 to 70 °C could inactivate endogenous antioxidant enzymes in the meat, resulting in rapid lipid oxidation. In this study, however, in the thigh muscle, TBARS values decreased at 65 °C for 6 h, which might be due to the additional reaction of oxidation products. Malondialdehyde produced by lipid oxidation is highly reactive [39] and reacts with compounds present in meat, such as amino acids [40]. In this regard, the decreased TBARS values in the thigh muscle with increasing cooking time were previously reported by Del Pulgar et al. [21]. Moreover, it is thought that this phenomenon, observed for only the thigh muscle, might be associated with a high concentration of myoglobin, potentially one of the initiators of lipid oxidation. The fact that the formation of metmyoglobin can accelerate the lipid oxidation of meat has been well documented [41]. In this regard, the thigh muscle contains more myoglobin than the breast muscle, and this would produce a more severe oxidative environment.

## 4. Conclusions

In this study, based on the thermal denaturation temperature of intramuscular connective tissue (55–60 °C), sous-vide cooking at 55 °C obviously decreased cooking loss and lipid oxidation but had no impact on the shear force. When compared to sous-vide cooking at 55 °C, sous-vide cooking at 65 °C could decrease the shear force but increase cooking loss. These tendencies of water-holding capacity and shear force at the two different temperatures were similarly observed for both chicken breast and thigh muscles. This result may confirm that the sous-vide cooking temperature for sufficient thermal denaturation of intramuscular connective tissue could be one of the critical factors in improving the tenderness of chicken meat. However, the increased sous-vide cooking temperature might eliminate the expected positive impact of sous-vide cooking on the water-holding capacity. To secure the advantages of both water-holding capacity and tenderness in sous-vide chicken, therefore, this study suggests that a novel strategy to apply together two temperature ranges based on the thermal denaturation of intramuscular connective tissue would be required.

## Figures and Tables

**Figure 1 foods-12-02592-f001:**
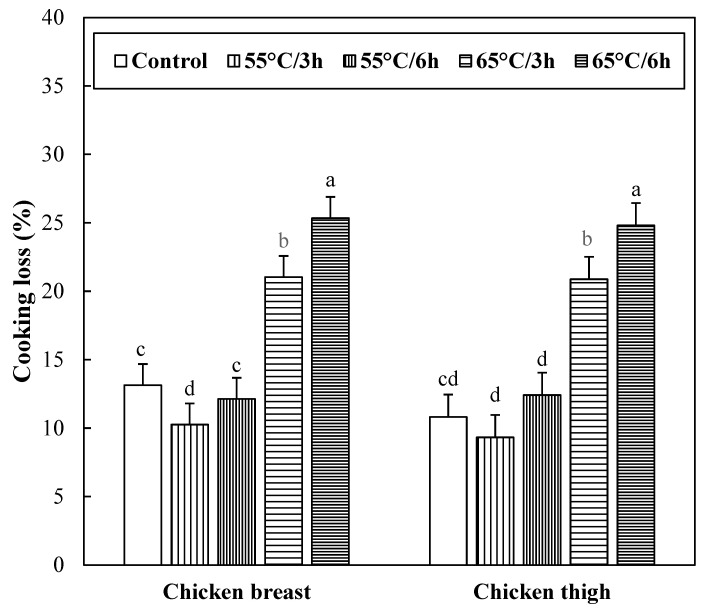
Cooking loss of sous-vide cooked chicken breast and thigh. Error bars refer to the standard error of the mean. a–d Means within the same muscle with different letters are significantly different (*p* < 0.05). Control samples were cooked at 75 °C when the core temperature reached 71 °C, sous-vide cooked samples were conducted at different conditions (temperature/time) under vacuum packaging.

**Figure 2 foods-12-02592-f002:**
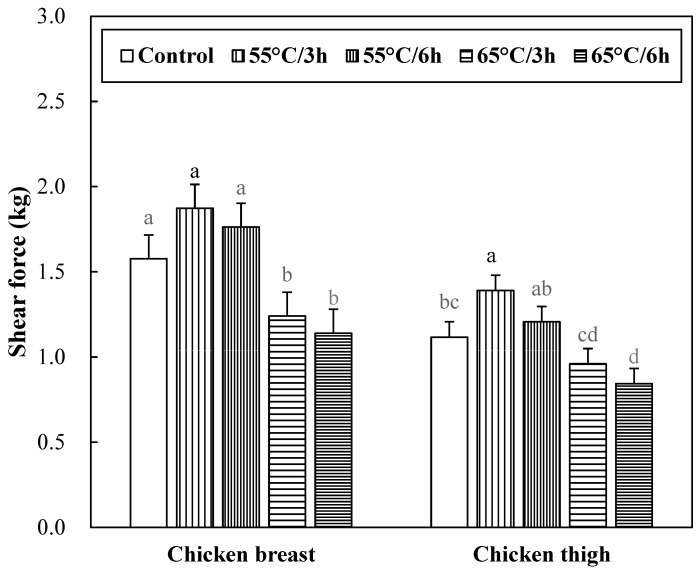
Shear force of sous-vide cooked chicken breast and thigh. Error bars refer to the standard error of the mean. a–d Means within the same muscle with different letters are significantly different (*p* < 0.05). Control samples were cooked at 75 °C when the core temperature reached 71 °C, sous-vide cooked samples were conducted at different conditions (temperature/time) under vacuum packaging.

**Figure 3 foods-12-02592-f003:**
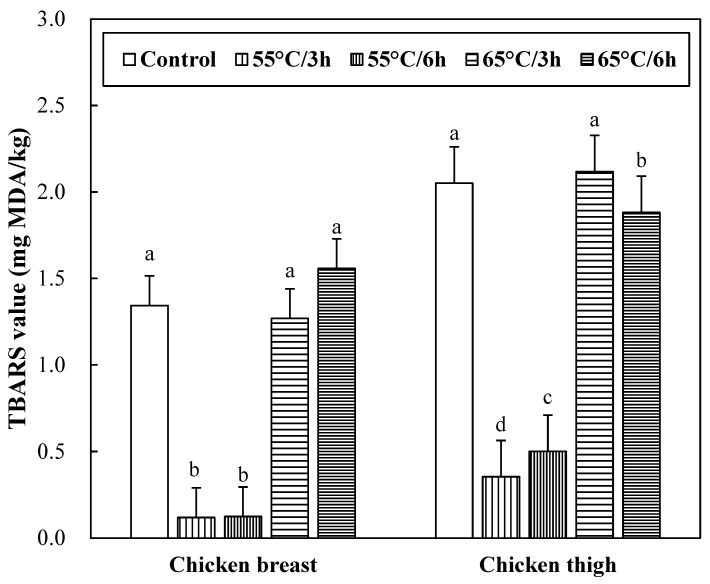
2-Thiobarbituric acid reactive substances (TBARS) of sous-vide cooked chicken breast and thigh. Error bars refer to the standard error of the mean. a–d Means within the same muscle with different letters are significantly different (*p* < 0.05). Control samples were cooked at 75 °C when the core temperature reached 71 °C, sous-vide cooked samples were conducted at different conditions (temperature/time) under vacuum packaging.

**Table 1 foods-12-02592-t001:** pH and color characteristics of sous-vide cooked chicken breast and thigh.

Effect	pH Value	CIE L* (Lightness)	CIE a* (Redness)	CIE b* (Yellowness)
*Muscle effect*				
Breast	6.07 ^b^	81.44 ^a^	3.39 ^b^	14.71 ^b^
Thigh	6.37 ^a^	75.14 ^b^	4.48 ^a^	15.35 ^a^
SEM ^(1)^	0.036	0.499	0.175	0.192
*Sous-vide effect*				
Control (71 °C)	6.29	79.63	4.00 ^b^	14.68 ^d^
55 °C/3 h	6.20	78.09	5.12 ^a^	13.65 ^e^
55 °C/6 h	6.25	77.53	4.23 ^b^	15.21 ^d^
65 °C/3 h	6.19	78.23	4.04 ^b^	14.92 ^d^
65 °C/6 h	6.17	77.98	2.29 ^c^	16.68 ^c^
SEM	0.057	0.788	0.277	0.304
*Muscle × Sous-vide interaction*				
Breast × Control	6.21	82.14	3.57	14.17 ^BC^
Breast × 55 °C/3 h	6.04	80.67	4.47	13.92 ^BC^
Breast × 55 °C/6 h	6.10	80.38	3.70	15.10 ^B^
Breast × 65 °C/3 h	6.01	81.04	3.60	14.97 ^B^
Breast × 65 °C/6 h	6.00	82.98	1.60	15.36 ^B^
Thigh × Control	6.38	77.11	4.42	15.19 ^B^
Thigh × 55 °C/3 h	6.36	75.51	5.78	13.38 ^C^
Thigh × 55 °C/6 h	6.40	74.69	4.76	15.31 ^B^
Thigh × 65 °C/3 h	6.36	75.42	4.47	14.87 ^B^
Thigh × 65 °C/6 h	6.34	72.99	2.98	17.99 ^A^
SEM	0.081	1.115	0.392	0.429
Significance of *p*-value				
Muscle effect	<0.001	<0.001	<0.001	0.028
Sous-vide effect	NS ^(2)^	NS	<0.001	<0.001
Interaction effect	NS	NS	NS	0.012

^(1)^ SEM: standard error of the means. ^(2)^ NS: non-significance (p ≥ 0.05). ^a,b^ Means within the same column with different letters between chicken muscles are significantly different (*p* < 0.05). ^c–e^ Means within the same column with different letters between sous-vide cooking conditions are significantly different (*p* < 0.05). ^A–C^ Means within the same column with different letters are significantly different (*p* < 0.05).

**Table 2 foods-12-02592-t002:** Moisture content, collagen content, protein solubility, and myofibrillar fragmentation index (MFI) of sous-vide cooked chicken breast and thigh.

Effect	Moisture Content (g/100 g)	Collagen Content(g/100 g)	Protein Solubility (mg/g)	MFI(Unitless)
*Muscle effect*				
Breast	71.66 ^a^	15.91 ^b^	43.57	117.83 ^a^
Thigh	70.00 ^b^	30.39 ^a^	49.92	23.36 ^b^
SEM ^(1)^	0.202	0.527	2.703	2.793
*Sous-vide effect*				
Control (71 °C)	71.15 ^d^	24.53 ^c^	43.64 ^d^	51.82 ^d^
55 °C/3 h	72.45 ^c^	24.34 ^cd^	63.74 ^c^	81.68 ^c^
55 °C/6 h	72.36 ^c^	23.21 ^cde^	57.50 ^c^	75.07 ^c^
65 °C/3 h	69.69 ^e^	22.08 ^de^	34.92 ^d^	72.31 ^c^
65 °C/6 h	68.49 ^f^	21.58 ^e^	33.93 ^d^	72.09 ^c^
SEM	0.320	0.735	4.274	4.416
*Muscle × Sous-vide interaction*				
Breast × Control	71.55	16.86	35.93	82.57 ^C^
Breast × 55 °C/3 h	72.77	16.65	57.26	140.81 ^A^
Breast × 55 °C/6 h	73.14	15.82	52.24	129.02 ^AB^
Breast × 65 °C/3 h	70.73	15.18	37.13	118.27 ^B^
Breast × 65 °C/6 h	70.12	15.01	35.28	118.46 ^B^
Thigh × Control	70.76	32.19	51.35	21.07 ^D^
Thigh × 55 °C/3 h	72.14	32.03	70.23	22.55 ^D^
Thigh × 55 °C/6 h	71.58	30.60	62.75	21.12 ^D^
Thigh × 65 °C/3 h	68.65	28.98	32.70	26.34 ^D^
Thigh × 65 °C/6 h	66.86	28.14	32.59	25.71 ^D^
SEM	0.453	1.039	6.045	6.245
Significance of *p*-value				
Muscle effect	<0.001	<0.001	NS	<0.001
Sous-vide effect	<0.001	<0.034	<0.001	0.002
Interaction effect	NS ^(2)^	NS	NS	0.003

^(1)^ SEM: standard error of the means. ^(2)^ NS: non-significance (*p* ≥ 0.05). ^a,b^ Means within the same column with different letters between chicken muscles are significantly different (*p* < 0.05). ^c–f^ Means within the same column with different letters between sous-vide cooking conditions are significantly different (*p* < 0.05). ^A–D^ Means within the same column with different letters are significantly different (*p* < 0.05).

## Data Availability

The data presented in this study are available on request from the corresponding author.

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
