# Peer review of "Physicochemical Properties of Chicken Breast and Thigh as Affected by Sous-Vide Cooking Conditions"

_foods, 2023, doi:10.3390/foods12132592_

Round 1
Reviewer 1 Report
This article is very interesting since it evaluates the effect of sous-vide cooking in some quality attributes of the most commercially used chicken muscles. As it was mentioned in the introduction, although the culinary benefits of sous-vide cooking have been well stablished, very little scientific research has been performed to understand and increase these benefits.
In general, the English of this manuscript needs to be improved, especially in the abstract and introduction section. Also, some asseverations made on the results were not supported by the statistical analysis (please see the attached file). I am attaching the revised file with some notes in order to quickly identify some of my concerns:
Line 202: Can the authors explain why the samples had a wide variability on a* in comparison to b* and L* values?
Line 228-229: Results on table don't show this behavior; actually, significantly opposite.
Line 231: Author should review and correct this statement.
Line 296: It was not significant.
Line 315: I suggest to authors to also indicate and discuss if a significant difference between breast and thigh muscles were found. Especially, why if thigh muscle had a significantly higher content of collagen, it has a lower shear force than breast?
Line 347: This only true for breast muscle, not for thigh
Line 366-365: if the interaction was significant, letters should be assigned throughout all treatments.
Conclusion: need to be improved

Although English is not my first language, I was able to notice some grammar mistakes that makes some of the statements difficult to understand, therefore, I suggest to properly edit the use of the language.
Author Response
Thank you for your valuable comment. Please see the attachment.

Reviewer 2 Report
The manuscript foods-2458062 entitled " Meat quality attributes of chicken breast and thigh as affected by sous-vide cooking conditions”
Overall, the paper is well written and very well explained.
I have few observations that need to be addressed to improve the paper’s quality.
Line 29: Please replace the words chicken breast, chicken thigh, sous-vide from keywords as these words are already available in the main title.
Introduction
Please add some data regarding the effect of this cooking technique to microbial analysis. Is it safe and effective to kill microbes at low temperatures such as 55 ºC. Because the authors did not study the effect of cooking method on microbial load, therefore, it is recommended that please refer some already published data on this topic.
Materials & methods
This section is by far too long. Thus, detailed description of well-known/published methods is not necessary. Please delete the detailed description of the following methods. The reference of already published method is enough or add only modification. No need to describe complete method.
2.3.3. Total collagen content
2.3.6. Total protein solubility
2.3.7. Myofibrillar fragmentation index
2.3.9. Shear force (Lipid oxidation)
Please replace shear force with lipid oxidation in section 2.3.9 of the materials and methods.
Results and discussion
Tables. Please provide standard deviation (SD) or standard error (SE) or mean standard error (SEM) in all tables.
Please replace superscripts with stars in tables.
Author Response

(The authors gave the same response as above.)

Reviewer 3 Report
In my opinion, the article should be revised.
However, relate to:
- Row 44: Please use °C. Please look for this aspect in the whole article.
- Row 353: Please use hours instead of hr.
- Please add more recent references.
Author Response

(The authors gave the same response as above.)

Reviewer 4 Report
While the work is well structured, detailed and supported by data, in my opinion it should have been set to evaluate the impact at very different cooking temperatures than those used, which with a very small difference ( 10°C) are meaningless.
From the extensive detailed bibliography on vacuum cooking it is now clear that a significant difference is in relation to more " brave" parameters around 40-45°C and less for times longer than 6 hours.
Therefore, the work does not bring any benefit to the current knowledge of vacuum cooking and meat characteristics.
Author Response

(The authors gave the same response as above.)

Reviewer 5 Report
Article with the title "Meat quality attributes of chicken breast and thigh as affected 2 by sous-vide cooking conditions" has 38 references cited on 12 pages. Relevant literature was used, the used quotation format corresponds to the desired pattern. Authors used 2 Tables and 3 Figures to present the results about the implications of sous-vide cooking conditions based on the denaturation temperature of intramuscular connective tissue on meat quality attributes of chicken muscles.
In my opinion, the article meets the basic requirements for Foods Journal, but I have some comments from the formal point of view. The scientific quality could also be improved if some information was better described. I miss the sensoric analysis with a sensory profile using descriptors. Subjective evaluation is for the meat quality essential.
The article has the title "Meat Quality Attributes ...", so if we admit that it could be without data from sensory analysis, it is necessary to adjust the name to the physico-chemical and structural parameters of chicken meat.
Line 77 - 84: "from a local poultry processing company and were freshly used for experiment within 1 day after purchase". Although the experiment was based on the monitoring of quality parameters, other results could be achieved, if the hybrid/breed factor, nutrition, welfare, etc. were included. Therefore, there is little information about the experimental material. I understand that all information cannot be obtained, yet it would be good to specify more material from a local poultry processing company.
Line 79: A comma was used, a decimal dot should be used correctly.
Lines 124 - 130: . Instrumental color measurement should should contain a reference reference with similar measurement procedure was used. Otherwise, it is necessary to describe why the Bone Side Surface was measured. Furthermore, what six random locations were used? Does it mean every time it was different? "Random"
Line 134: There is an extra bracket.
Lines 157 and 164: 2.3.8. and 2.3.9. the same title of the chapter: Shear force (correctly TBARS)
I have serious reservations about the brief names of chapters in the results and discussion. 3.3. Total collagen content 3.4. Total protein solubility 3.5. MFI 3.6. Shear force (If the reader opens the article in the middle, he does not know what it is here and where it belongs.)
To summarize the known facts, the article does not differ from other articles, but lacks a clear goal and it is not clear what the authors wanted to publish as essential information.
Author Response

(The authors gave the same response as above.)

Round 2
Reviewer 4 Report
I appreciate the response and after reading the many revisions I think the work makes an interesting contribution.
Reviewer 5 Report
I have no other comments.